# An Ecological Model for High-Risk Professional Decision-Making in Mental Health: International Perspectives

**DOI:** 10.3390/ijerph18147671

**Published:** 2021-07-19

**Authors:** Cheryl Regehr, Guy Enosh, Emily Bosk

**Affiliations:** 1Factor-Inwentash, Faculty of Social Work, University of Toronto, Toronto, ON M5S 1A1, Canada; 2School of Social Work, University of Haifa, Haifa 3498838, Israel; enosh@research.haifa.ac.il; 3School of Social Work, Rutgers University, New Brunswick, NJ 08901-8554, USA

**Keywords:** mental health, social worker, risk assessment, child welfare, decision-making

## Abstract

Mental health professionals are frequently presented with situations in which they must assess the risk that a client will cause harm to themselves or others. Troublingly, however, predictions of risk are remarkably inaccurate even when made by those who are highly skilled and highly trained. Consequently, many jurisdictions have moved to impose standardized decision-making tools aimed at improving outcomes. Using a decision-making ecology framework, this conceptual paper presents research on professional decision-making in situations of risk, using qualitative, survey, and experimental designs conducted in three countries. Results reveal that while risk assessment tools focus on *client factors* that contribute to the risk of harm to self or others, the nature of professional decision-making is far more complex. That is, the manner in which professionals interpret and describe features of the client and their situation, is influenced by the worker’s own personal and professional experiences, and the organizational and societal context in which they are located. Although part of the rationale of standardized approaches is to reduce complexity, our collective work demonstrates that the power of personal and social processes to shape decision-making often overwhelm the intention to simplify and standardize. Implications for policy and practice are discussed.

## 1. High-Risk Professional Decision-Making in Mental Health

Mental health legislation, and the policies of organizations serving those with mental health challenges, generally attempt to balance three potentially competing principles: the civil liberties of individuals to live as they choose; the responsibility of society to ensure the safety of those who may be unable to appreciate the nature and consequences of their actions and thus serve as a risk to themselves; and the responsibility of society to ensure that the behavior of one individual does not compromise the safety and security of others [1]. While legislation and policies set direction and create a framework for operationalizing these principles, ultimately, society and organizations rely on the decisions of frontline professionals to interpret legislation and policy and determine how they should be applied in a particular case. The complex situations confronting mental health professionals must often be distilled to: by virtue of their current mental health status and circumstances, does this person represent a risk of harm to themselves or others at this time or in the near future?

Troublingly, predictions of whether an individual will ultimately harm themselves or others are remarkably inaccurate even when assessments and decisions are made by professionals who are highly skilled and highly trained [2,3]. The consequences of being wrong not only result in the tragic loss of human life or significant injury but also public and political outrage [4,5]. Munro suggests that three processes then emerge: punish those who make incorrect predictions; reduce the role of human reasoning in decision-making; and increase monitoring [4]. In this vein, many jurisdictions across the world have moved to standardize professional judgment through the use of decision-making tools in a diverse range of clinical situations, including those involving recidivism of mentally ill violent offenders [6,7], suicide risk [8,9], and risk of child abuse [4,10]. The assumption is the decision-making tools will result in increased safety and better public health outcomes. Another rationale for the use of risk assessments is to redress implicit bias (decisions grounded in unconscious prejudices) [11]. By establishing objective criteria on which to make decisions, these tools are intended to eliminate subjective assessments that lead to unfair outcomes.

Nevertheless, there is considerable evidence that the manner in which professionals use decision-making tools and approach risk assessment is highly variable, resulting not only in differing judgments amongst professionals assessing the same cases [12,13] and between professionals and standardized tool scores [13,14] but also by the same professional at different times [15]. Understanding the nature of these differences in decision-making is critical to addressing the challenge of risk of harm to self and others arising from mental illness and distress. This paper presents the work of three scholars in three different parts of the world (Canada, Israel, and the United States), conducting research on professional decision-making in high-risk situations involving risk of harm to children at the hands of their parents, and involving clients contemplating suicide. We include research involving social workers in child welfare in addition to those in mental health due to the high incidence of mental illness and addictions in parents assessed to be at risk of harming their child [16,17], and the impact of early experiences of maltreatment to adult mental health [18]. While the authors have not previously worked together, we have discovered a remarkable synergy in our work that has independently and collectively concluded that while risk assessment tools focus on *client factors* that contribute to the risk of harm to self or others, the nature of professional decision-making is far more complex. Although part of the rationale of standardized approaches is to flatten complexity, our collective work demonstrates that the power of personal and social processes to shape decision-making often overwhelm their intention to simplify and standardize. 

Ecological theory posits that human behavior is a consequence of the interaction between the individual (personal characteristics and formative history); the person’s immediate context (family, peers, and cultural links); the environment in which the person resides (including the resources and limitations of their neighborhood and community); and macrosystem factors (those that are economic, social, and political in nature) [19,20]. This model has been applied to personal decision-making in a range of areas, such as disclosing intimate partner violence [21] and undertaking behaviors that increase or decrease the risk of contracting HIV [22]. Baumann and colleagues expanded upon ecological theory and proposed the concept of decision-making ecology for professionals assessing the risk of child abuse. This decision-making ecology includes the contributions of case factors, organizational factors, external factors, and decision-maker factors [23]. 

This conceptual paper presents an empirically-based theoretical reformulation of risk assessment and suggests practical policy interventions that arise from this reformulation. Specifically, we offer a modification of the ecological decision-making model in order to apply it to professional decision-making in mental health risk assessment. To do so, we integrated the work of three separate researchers in three different cultural and social systems, using varied methodologies (quantitative and qualitative; experimental and survey-based), as cumulative and triangulated data sources. In this revised model, we contend that the client factors which form the foundation of risk assessment tools are interpreted, evaluated, and weighed within the context of the individual worker’s personal and professional experience, and the organizational dynamics, policy framework, and societal context in which the work is conducted. Ultimately it is these combined factors that contribute to any professional decision regarding impending risk, as depicted in Figure 1 below:

## 2. Client Factors and Decision-Making

Client factors are at the heart of models to improve decision-making in mental health, such as through the use of actuarial or structured professional judgment tools. For instance, actuarial risk assessment tools use statistical algorithms to predict specific risk outcomes based on large data sets consisting of individuals who have committed the acts in question (such as suicide attempts or harming others) [24,25]. A particular cluster of characteristics results in an increased statistical probability that the individual will commit a violent act. This then results in a risk score, which, in turn, may be associated with specific outcomes or treatment approaches. For instance, in child welfare, a particular score may predict that a parent presents a safety risk to a child to the extent that out-of-home placement is warranted [14]. While from a policy perspective, this approach is appealing and intuitively one might expect predictability in outcomes, our collective research has demonstrated that even when client factors are held constant through the use of written vignettes or standardized clients who provide consistent responses, professionals arrive at highly divergent conclusions [13].

Most disturbingly, in situations where the level of uncertainty is high, workers may become more reliant on personal biases and personal experiences—both developmental and professional—in their appraisals. Consequently, client characteristics related to identity and socioeconomic status may contribute to the appraisal of risk [12]. In order to investigate this, a series of experimental design studies were conducted in Israel with case workers (N = 105), community health professionals (N = 412), and members of planning and decision committees (equivalent to family case-conferences), consisting mostly of social workers (N = 290, in 50 committees across the country). Life-like vignettes (originally drawn from actual case records) were manipulated in order to assess the possible biasing effects on professional decision-making of factors, such as client’s ethnic identity, socioeconomic status [12,26], religiosity [27], or gender [28]. Overall, each of these client characteristics was found to affect workers’ professional decision-making, especially in cases of higher ambiguity. Specifically, results demonstrated that those assessed to be at the highest risk were male, members of ultra-orthodox religious communities, and members of ethnic minorities, even when standard risk factors were held constant. Thus, overlaying risk decisions are powerful ideas that are based on longstanding social histories of perceiving racial and ethnic minorities to be ‘more risky’. While one rationale for the use of actuarial-based tools is to circumvent implicit bias by standardizing the factors on which to base decisions, this line of research suggests that these tools are ultimately an ineffective means of eliminating these perennial challenges in interpreting case factors and in fact simply mask bias [29,30].

Naylor has suggested “A rigorous decision analysis can indeed delineate for typical patients the available options, clinical consequences, and any associated degree of uncertainty. But to paraphrase Voltaire, typical patients are far from typical” [31] (p. 523). In an actual clinical encounter, professionals rapidly consider a wide range of contextual factors that they incorporate into their decisions [32]. In assessing risk vignettes in our studies, these factors may be assumed based on previous experiences of ‘typical patients’ with these characteristics, referred to in medical research as ‘illness scripts’ [33]. These scripts are formed through exposure to a wide range of clients over the course of a professional career. Typical scripts may also serve as a baseline for assessing divergence from ‘typicality’ and suspecting that something is wrong or of concern [34]. Describing decisions to report suspected child abuse for investigation in a US study, professionals indicated that they were not necessarily looking for red flags but rather sensed an incongruence in the case presented, an incongruence that was based on implicit scripts [30].

## 3. Professional Experience Factors and Decision-Making

Professional experience encompasses a wide range of factors, including the educational background of the individual, specific training in topics, such as suicide risk assessment, acquired expertise through years of practice, and, unfortunately, negative professional experiences that may influence perceptions [35]. Negative experiences that can lead to significant levels of distress in professionals, including symptoms of post-traumatic stress, may involve threats or physical harm to the professional [36,37]; repeated exposure to traumatized and highly distressed individuals [38,39]; and the tragic loss of a client, for instance to suicide [40,41].

An experimental design study conducted in Canada sought to determine the degree to which previous work-related experiences and consequent emotional states of clinicians influenced professional judgment regarding acute risk in clients presenting with suicidal ideation [13]. Seventy-one experienced social workers and social work students conducted suicide risk assessments on two standardized patients, one depicting an adolescent in a situational crisis and one depicting a chronically depressed woman. Results revealed considerable variability in clinical decision-making, with approximately one-third of participants judging that the adolescent should be hospitalized due to acute risk of suicide, and two-thirds judging that the woman should be hospitalized due to acute suicide risk; the remaining participants judged that the clients would be safe without hospitalization. Decisions about whether to hospitalize were predicted by scores workers ascribed on standardized risk assessment measures—however, these ascribed scores were highly variable. A significant proportion of the variability in judgments of safety on standardized risk assessment measures could be attributed to the pre-existing emotional state of the professional. In both simulated clinical cases, *higher* levels of post-traumatic stress symptoms in the social workers, as measured by scores on the Impact of Event Scale—Revised [42], were associated with *lower* assessed suicide risk as measured by scores on standardized suicide risk assessment measures [9,43]. This finding replicated that of an earlier study with child welfare social workers in which increased levels of post-traumatic symptoms reduced the likelihood that a worker would determine that a child was at risk of abuse [44].

Similarly, a quantitative standardized vignette-based study regarding the assessment of risk and mandatory reporting conducted in Israel with 412 healthcare professionals found perceived workload stress had a significant influence on assessment and decision making. While the burden of actual workload may be open to contest, whether it is reasonable, too heavy, or a bargaining position in union negotiations, there are indications that perceived workload stress, as measured by questions such as “How often does your job leave you with little time to get things done?” “How often do you have to do more work than you can do well?”, increases the tendency of healthcare professionals to assess risk as higher [26].

## 4. Personal and Interactional Experiences as Influences on Decision-Making

Another way in which client factors are filtered by the professional making the high-risk decision is relational. For instance, in a qualitative interview-based Israeli study, 18 healthcare professionals from deliberately varied and heterogeneous community clinics reported that concerns that decisions would lead to a breakdown in the working relationship with the client affected the manner in which they appraised risk, particularly if they were members of smaller, close-knit communities [34]. In qualitative interviews following participation in simulated risk assessments with standardized patients in Canada, social workers also reported that relational factors influenced perceptions of risk and their confidence in their risk appraisal. That is, if a relationship could be established between the professional and the client, this was viewed as lowering the overall risk of future harm [45].

Providers’ own trauma histories and relational experiences also influence their risk assessment. In a study of 271 US mental health providers at three separate agencies implementing Trauma-Informed Care (TIC), those with a higher rejection sensitivity (a relational construct grounded in histories of complex trauma) were less likely to be open to the principles of TIC [46]. Adopting TIC required shifting their assessments of clients from behavior-based to socio-emotional and tolerating higher levels of risk. These providers were also more likely to report an intent to leave their jobs when their organizations began adopting trauma-informed interventions. In a similar vein, a study in Israel involving 290 members of multidisciplinary child protection teams compared individual decision-making to group decision-making using standardized vignettes and found complex interactions between previous exposure to client aggression and assessments of risk [47,48]. For instance, exposure to verbal violence or vandalism of personal property *decreased* the probability that individual workers would assess a client to be at high risk of violence and that a child required out-of-home placement. However, accumulated exposure to threats towards committee members *increased* the probability of a recommendation by the committee for out-of-home placement. These relational factors sit outside of standardized assessment tools.

## 5. Organizational Influences on Decision-Making

Organizations in which social workers are employed are buffeted by financial constraints, increased client demand, and changing government policies and priorities. Time spent with clients is replaced by accountability reporting resulting in increased demand and decreased control [23,49,50,51]. Organizational influences can include the availability of resources. For instance, the reduction in inpatient beds for those experiencing acute episodes of serious mental illness may limit the choices available. Thus, risk tolerance and decisions to avoid hospitalization and maintain the individual in the community may be based on an awareness that finding a bed will be exceedingly difficult.

Organizational influences also include team dynamics. In a pilot intervention aimed at improving decision-making in situations of high-risk, Canadian professionals (social workers, nurses, and occupational therapists) in a large, urban mental health facility participated in a master class series for practicing professionals which was paired with explicit attention to biological, emotional, cognitive, and environmental influences on decision-making during real world practice experiences, and opportunities for self-reflection following simulated assessment. Participants were asked to track their decision-making in real-life clinical encounters and record their cognitive processes, emotional reactions, and, via the use of wearable technology, monitor their heart rate response [52]. While client situations such as meeting a distressed family member, threats of violence against the worker, and assessing suicide risk were associated with elevated physiological and emotional response, team dynamics significantly contributed to stress and, ultimately, the decisions made. As one participant noted in a debriefing session, “One consideration is my tolerance for risk vs. the team’s tolerance for risk. I often have a physiological response relating to anticipation of the team’s pushback.” Others spoke about the politics of decision-making and the ability of a professional to voice their own unique assessment based on their position in the team hierarchy. “The politics of these big decisions that one would make in a work setting… and kind of where you stand in the grand scheme of things. Are you up top on the hierarchy or the bottom because you feel less empowered and kind of uncomfortable voicing your beliefs if you’re on the lower end of the scale?” Thus, individual decisions regarding risk are shaped by team culture and politics.

Furthermore, the structure and internal dynamics of the decision-making team may have major impacts on the outcomes of team decision-making processes. In the study of 50 treatment-planning committees for children at risk in Israel using standardized vignettes, it was found that the structure of the committee, as presented by the presence or absence of specific authority figures, affected the decision outcome [48]. More specifically, the presence of a specially trained child-protection social worker (serving also as the link between the child welfare system and the court) significantly increased the likelihood that the decision would include a court involvement. Conversely, the presence on the committee of an adoption social worker increased the likelihood for an out-of-home placement decision significantly.

## 6. Policy Influences on Decision-Making

Beginning in 1980, Lipsky argued that “the decisions of [individuals whom he called] street level bureaucrats, the routines they establish, and the devices they invent to cope with uncertainties and work pressures, effectively become the public policies they carry out” [53] (p. xiii). This formulation suggests a wide breadth of discretion within which the professional operates. Indeed, despite legislative reform in Canada that has limited the conditions under which an individual can be admitted to a psychiatric hospital against their will, research has demonstrated that rates and reasons for involuntary admission have remained relatively stable over the decades [54]. It has been suggested that despite changes, judges and doctors have continued to use a ‘common-sense’ perspective to assess the risk of harm [55] and respond to the lived experience of individuals with mental illness and their families [18].

Research conducted in the US demonstrated that organizational policy also shapes the conditions under which workers use risk assessments [14]. An in-depth comparative case study examined the influence of different policies regarding the use of the same assessment tool for determining the risk of child maltreatment in two State child welfare agencies in the US [14]. One state required that the risk score determine the next step for cases, while the other state simply suggested that the risk score should inform decision-making [56]. When the risk score only informed decision-making, few workers reported using this information in any meaningful way, making it practically useless for enhancing risk assessment. In the state that relied on the risk score, the majority of workers reported that they felt they often had to take case actions with which they strongly disagreed, leading to feelings of distress. Both organizational policies presented different risks: in one state, an overreliance on risk scores meant case outcomes may be too punitive, while an under-reliance on risk assessments may mean that important case factors are being left out of decision-making.

In this study, workers tended to disagree with the risk score when they believed it to rely too heavily on demographic factors, such as the number of children in the home, previous anonymous complaints of child abuse that have not been substantiated, and the parent’s own personal history of childhood sexual abuse, rather than nuanced case factors. While clinical judgment is supposed to be a ‘check’ on algorithms, in this case, an organizational policy that tied the risk score to case trajectories with no provision for worker overrides made it impossible to balance actuarial knowledge with clinical knowledge. The social power of numbers to present an ‘objective finding’ combined with organizational imperatives to manage risk means that when it comes to high-stakes decisions, it can be difficult for frontline workers to challenge risk scores because of the ways these challenges may put the organization itself ‘at risk’.

## 7. The Societal Context and Professional Decision-Making in Mental Health

Social work risk assessment frequently occurs within policy frameworks and organizational procedures that are highly structured and prescriptive in reaction to previous negative outcomes. Social workers approach risk situations aware of the risk of public outcry, personal liability, and organizational sanctions [49,57]. In a medical context, Leung and colleagues suggested that additional factors that may influence professional judgment include those that are avowed (those that align with professional values, such as client best interest); those that are unavowed (including organizational pressures and policies), and those that are disavowed (such as personal liability) [58]. This can result in decisions that reflect “defensive practice by professionals so that [client] best interests are not always at the heart of the decisions” [50].

The research that we have undertaken highlights the impact of these factors on professional decision-making in high-risk situations. For instance, in a qualitative study of 18 healthcare professionals regarding mandatory reporting of child abuse in Israel, participants shared the costs of mistakes not only for the child and family but also for themselves. One professional dreaded “leaving a child in danger” and later “seeing his picture in the newspaper” [34] (p. 101). In the pilot study on improving decision-making in Canadian mental health practice, a participant similarly spoke about choosing the least negative amongst imperfect options in part in fear of the negative consequences of being wrong. “As a clinician, the easier thing is to push for hospitalization. Then I don’t have to worry, and that risk doesn’t exist for a period of time. But the overall impact of that on the client is quite damaging” [52]. These findings are echoed in work from the US where child welfare workers reported that they value using risk assessments because they protect them from the devastating personal and professional consequences of making mistakes when lives are at stake [59]. This approach, however, means that the costs of mistakes that overestimate risk are not born by the system itself, leaving little incentive to manage risk for clients in ways that may be less damaging to them [30].

As noted above, in a situation of high uncertainty, professionals can become more reliant on bias to shape their assessments. One rationale for standardized risk assessments is to neutralize that bias by only relying on objective indicators. Less attention has been paid to social factors that shape choices to either follow or ignore the recommendations of risk assessments. Returning to the US child welfare example, in the state that used risk assessment scores to determine case outcomes, one-third of workers decided to ‘alter’ the risk score so that the case trajectory would align with their clinical judgment. Only white and male participants, however, reported that they chose to violate policy even though an equal number of racial and ethnic minorities noted equally strong disagreements with the risk score [53]. Instead of being an equalizing force, standardized processes in certain organizational contexts can act as another site for inequality in who experiences control over their decisions.

Finally, risk assessment in mental health focuses on individual risk, history, and behavior. People’s risks, though, are shaped by social environments, which, in turn, are shaped by policy [58]. Is lack of housing most accurately understood as an individual risk or a reflection of policy choices that shape the social environment? Whatever the answer, most risk assessments treat social conditions as individual risks. Risky social conditions instead become understood as risky individuals, limiting our understanding of other areas that may need intervention and attention, such as public funding for childcare, housing support, or other basic income supports, that are tied to lower risk across areas of mental health and wellbeing.

## 8. Summary and Implications

The various research projects reported in this paper independently and collectively demonstrate that the decision-making of social workers and other mental health professionals faced with high-stress risk assessment is subject to a multiplicity of forces and influences that extend well beyond the client factors included in standardized risk assessment tools. The three countries in which we have conducted our research, Canada, Israel, and the United States, have different political systems, different organizational structures for the provision of mental health and child welfare services, different cultural influences, and different groups of people who face systematic barriers to success and inclusion. Yet despite these differences in context, pre-existing biases, organizational pressures, imperfect social policies, and societal pressures on workers have similar influences. Each of these factors, some of which are unavowed and others of which are disavowed [58], contribute to decision-making practices that override, shape, or ignore the factors included in standardized tools. Despite attempts to standardize, workers apply their own personal and professional judgments to assessing risk.

The implications of the combined model and findings from three different international contexts presented here call for interventions and change on two separate but related levels, individual and organizational/policy. First, on the personal level of the individual social worker, it becomes apparent that personal issues play a paramount role in decision-making. Personal assessments of risk are biased by implicit attitudes, by the personal history of the worker, by interpersonal experiences at work, as well as by work relationships of the worker with colleagues and with the organization. As was described above, the use of standardized instruments and systems of assessments have proved to fail, and workers indicated that they had redefined the scale or the case in order to fit their personal judgment. Thus, using external constraints, such as actuarial systems of risk, may not be a solution. We suggest that the solution should come from within. Workers should be educated and encouraged to actively engage in self-reflective practices in order to heighten awareness of their own biases. However, abundant research in social psychology has indicated that individuals are perfectly capable of detecting the personal biases of others while are usually impervious to their own faults [60]. How can we overcome such hindrances? One way to approach this would be to use dialectical reflectivity [61,62]. In practice, this would entail actively searching for contra-indications for one’s assessments, judgments, and decisions. The active engagement in dialectical reflectivity can and should be incorporated into formal training of workers, as well as into the assessment process. In other words, we should teach workers about biases and how dialectical thinking—the active search for negation, for inconsistencies, for apparent contradictions, can serve as debiasing processes.

Nevertheless, given the inherent human tendency to see the mote in another’s eye and not the beam in one’s own, this then suggests that organizational and policy level changes are necessary. We suggest that structured decision-making tools should not be constructed of specific signs or measures that attempt to “replace the expert” or which plug in specific factors to reach a predetermined outcome [63]. Rather we support the development and use of evidence-based guidelines or frameworks that allow flexibility in incorporating case-specific and contextual factors by asking directive questions that help the workers and teams engage in debiasing [11] through dialectical reflectivity. This process-based approach would structure integration of known factors that elevate risk alongside a focused process for applying these insights to the case itself. Instead of seeing the decision-maker as separate from how data are interpreted, a process-based approach invites thoughtful discussion about these areas [52]. Data from three different countries suggest that risk assessments are not able to standardize decisions as fully as intended. Instead, new models are needed that attend to the ways in which risk assessment is always socially situated, shaped by intersecting social, organizational, personal, and client level factors. Engaging in structured discussion and dialectical reflexivity invites the decision-maker to explicitly make connections between these domains rather than to assume that they will be able to be flattened out or eliminated.

## 9. Conclusions

Mental health professionals are frequently presented with situations in which they must assess whether a client by virtue of their mental health status and circumstances presents an imminent and serious risk of harm to themselves or others. To avoid the tragic outcomes of failing to identify a high-risk situation, researchers have designed, and organizations and jurisdictions have adopted risk assessment tools that aim to standardize and improve professional judgment. These tools primarily focus on client factors. However, they fail to address the manner in which professionals interpret these factors and the manner in which personal and processes shape decision-making processes. Applying and modifying an ecological decision-making model to mental health practice, we contend that client factors are interpreted, evaluated, and weighed within the context of an individual worker’s personal and professional experiences, the relationship between the worker and the client, the organizational dynamics, policy framework, and societal context in which the interaction occurs. Improving risk assessment in mental health does not rest on ridged tools but rather interventions that support professionals to examine their own decision-making processes, and organizational policies and procedures that provide the scope for workers to do so.

## Figures and Tables

**Figure 1 ijerph-18-07671-f001:**
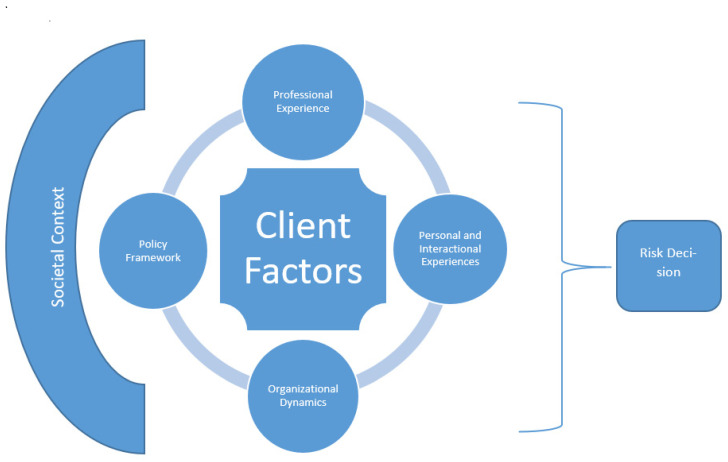
An Ecological Model for High-Risk Professional Decision-Making in Mental Health.

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
