# Peer review of "An Ecological Model for High-Risk Professional Decision-Making in Mental Health: International Perspectives"

_ijerph, 2021, doi:10.3390/ijerph18147671_

Round 1
Reviewer 1 Report
This is a great, thorough paper that presents a great synthesis of the ecological factors surrounding risk assessment which is much needed within the mental health sector. The only change I would suggest is some more detail in your discussion of "evidence-based guidelines or frameworks". This sounds like an important recommendation and it would be good to have some more detail about what this would look like in practice and whether there are any good examples of this approach currently being used.
Author Response
July 1 2021
Professor Gavin Davison, Professor Lisa Brophy, and Professor Jim Campbell Guest Editors
International Journal of Environmental Research and Public Health
Dear Gavin, Lisa and Jim
Thank you very much for the invitation to submit an article to the Special Issue on International Perspectives on Mental Health Social Work of IJERPH. My co-authors and I are pleased to be invited to revise and resubmit the manuscript.
We are pleased that reviewer #1 indicated “This is a great, thorough paper that presents a great synthesis of the ecological factors surrounding risk assessment which is much needed within the mental health sector.” Reviewer #2 indicated the problem identified in the paper is urgent and important and identified that the international composition of the authors, the use of mixed methods, and the selection of decision-making ecology as a theoretical perspective were strengths. However, the second reviewer identified some areas that could be strengthened. We address these concerns below and highlighted the changes in the revised manuscript.
- R#1 has asked us to provide additional information regarding our recommendation for “evidence-based guidelines”. To this end we have added a paragraph in the discussion section.
- R#2 suggestions that we needed a clearer goal, hypothesis and research question and that the article type is undefined. To this end, we have added a paragraph explaining that this is a paper focused on the development of theory based on our research and is not aimed at presenting a singular research project.
- R#2 has asked for additional information on participants, techniques and research results. Given the space constraints and the theoretical nature of the current paper, we have added information to the studies presented, but not to the degree we would if focusing on present new research or if we were not constrained by length.
- R#2 has requested additional analysis of our international perspectives. To this end, we have included a paragraph at the beginning of the Summary and Implications section that highlights the similarities in our findings despite high variable contexts and expanded on the implications these shared findings suggest.
We believe that these suggested revisions have served to strengthen the manuscript and look forward to your comments.
Reviewer 2 Report
The presented article is devoted to an undoubtedly important and urgent problem on high-risk professional decision-making in mental health. The urgency and practical significance of this problem has increased in the COVID-19 pandemic situation, when the level of stress has sharply increased both among social workers and their clients.
The strengths of the article are: (1) the international composition of the authors (Canada, Israel, and US), which allows them to consider the problem from an international perspective, as well as (2) the use of a mixed methodology (qualitative, survey, and experimental designs). The advantage of present research is also the use of a decision-making ecology framework.
However, in our opinion, the authors did not manage to embody these strengths in the presented article.
The weaknesses of the article are as follows:
- Lack of a clearly formulated goal, hypothesis and research questions;
- Undefined article type: is it a research article or a review article? In the first case, there is a lack of a detailed description of the participants, techniques, research results of each of the mentioned research designs (qualitative, survey, and experimental). In the second case, more analyzed sources and references, and clearer research questions are required;
- Although the authors claim an international perspective of their research, the article has no cross-cultural analysis and does not clearly highlight similarities and differences in decision-making among social workers in different countries.
Thus, despite the relevance and significance, this article does not meet the requirements for scientific publications and cannot be published as presented.
Author Response

(The authors gave the same response as above.)

Round 2
Reviewer 2 Report
I would like to re-emphasize the relevance and importance of the problem discussed in this article. I thank the authors for carefully considering my questions and comments and taking them into account in the new version of the article.
I am satisfied with the additions and corrections that the authors made to the article. The only thing I can suggest to add the word "Model" in the title of the article and in the caption to the figure:
The Ecology Model of High-risk Professional Decision-Making in Mental Health: International Perspectives / The Ecology Model of High-risk Decision-Making in Mental Health
In this case, the reader will immediately understand the theoretical nature of this article.
Summarizing, I recommend this article for publication. Maybe it needs a little editing, including English, but this is up to the Editor’s decision.
Author Response
The title has been changed as recommended.